# Adsorption Characteristics of Activated Carbon Fibers in Respirator Cartridges for Toluene

**DOI:** 10.3390/ijerph18168505

**Published:** 2021-08-12

**Authors:** Jo Anne G. Balanay, Jonghwa Oh

**Affiliations:** 1Environmental Health Sciences Program, Department of Health Education and Promotion, College of Health and Human Performance, East Carolina University, Greenville, NC 27858, USA; 2Department of Environmental Health Sciences, School of Public Health, University of Alabama at Birmingham, Birmingham, AL 35233, USA; jonghwa@uab.edu

**Keywords:** activated carbon fiber, breakthrough, toluene, respiratory protection, service life, volatile organic compound

## Abstract

Respirator use has been shown to be associated with overall discomfort. Activated carbon fiber (ACF) has potential as an alternative adsorbent for developing thinner, lightweight, and efficient respirators due to its larger surface area, microporosity, and fabric form. The purpose of this pilot study was to determine the adsorption characteristics of commercially available ACF in respirator cartridges with varying ACF composition for toluene protection. Seven ACF types (one cloth, six felt) with varying properties were tested. Seven ACF cartridge configurations with varying ACF composition were challenged with five toluene concentrations (20–500 ppm) at constant air temperature (23 °C), relative humidity (50%), and air flow (32 LPM). Breakthrough curves were obtained using photoionization detectors. Breakthrough times (10%, 50%, and 5 ppm) and adsorption capacities were compared among ACF cartridge configurations to determine their suitable application in respiratory protection. Results showed that ACF cartridges containing the densest ACF felt types had the longest average breakthrough times (e.g., ~250–270 min to reach 5 ppm breakthrough time) and those containing ACF felt types with the highest specific surface areas had the highest average adsorption capacity (~450–470 mg/g). The ACF cartridges demonstrated breakthrough times of <1 h for 500 ppm toluene and 8–16 h for 20 ppm toluene. The ACF cartridges are more reliable for use at low ambient toluene concentrations but still have potential for use at higher concentrations for short-term protection. ACF felt forms with appropriate properties (density of ~0.07 g/cm^3^; specific surface area of ~2000 m^2^/g) have shown promising potential for the development of lighter and thinner respirators for protection against toluene.

## 1. Introduction

Millions of workers in various workplaces throughout the United States are required to use respirators for protection against airborne pollutants that may cause diseases or death [1]. The proper use of respirators may prevent a significant number of deaths and illnesses annually. Granular activated carbon (GAC) is the current standard adsorbent in respirators against various gases and vapors, including volatile organic compounds (VOCs), because of its low cost, efficiency, and available technology. However, due to its granular form, GAC needs containment (i.e., in a cartridge or canister), which contributes to the weight and bulkiness of air-purifying respirators, in addition to the weight of the GAC itself [2]. Respirator use has been shown to be associated with overall discomfort [3,4]. Lightweight respirators are significantly rated as more comfortable than heavy ones [5] and, thus, are more likely to be worn. Therefore, it is essential to develop improved respirators to address such discomfort-related and other issues, thereby improving user compliance and protecting worker health.

Activated carbon fibers (ACFs), which are obtained from the carbonization and activation of polymeric fibers from various precursors [6], have been considered as good alternative adsorbents for developing thinner, lightweight, and efficient respirators for VOC protection because of their larger surface area, higher adsorption capacities, lower critical bed depth, greater number of micropores, lighter weight, and self-containing fabric form compared to the GAC [7,8,9,10]. The small fiber diameter of ACFs allows homogeneous fiber activation, leading to the ACF’s narrow pore size distribution in the micropore range [11,12]. Unlike the more complex porous network of GAC (i.e., macropores branching to mesopores then to micropores), ACF has micropores that are directly accessible from the fiber surface by adsorbates, resulting in faster adsorption kinetics [6,7,8]. ACFs were demonstrated to effectively capture VOCs from gas streams at a broad range of concentrations [13,14,15,16,17]. Studies investigating the use of ACF in sorbent applications have been ongoing for a few decades, particularly for air pollution control of VOCs [18,19,20,21,22,23,24,25]. ACF has promising potential for short-term respiratory protection of workers and public health during accidental or intentional release of toxic air pollutants in a catastrophic event [9], and during routine work. However, the application of ACF as an adsorbent in respiratory protection remains understudied.

Our previous ACF research studies have investigated the adsorption characteristics of ACF materials in two forms (i.e., woven cloth and non-woven felt) and compared them to GAC by challenging the carbon adsorbent samples with 500 ppm toluene, as a VOC representative, in a sample chamber [9]. Results have shown that ACF types with surface areas comparable to that of GAC have higher adsorption capacity and lower critical bed depth [9]. We further investigated the adsorption characteristics (i.e., adsorption capacity, critical bed depth, and breakthrough) of the same ACF materials by challenging the adsorbent samples with toluene at six different concentrations (50–500 ppm) in the same sample chamber, focusing on comparing the ACF cloth with felt [2,26]. Results showed that ACF cloth, due to its denser form, had a higher adsorption capacity, longer breakthrough time, and lower critical bed depth than the ACF felt with similar specific surface area, suggesting the potential application of ACF cloth for thinner and lighter respirators [2,26]. To further investigate the application of ACF in respirator components, various ACF materials (one cloth, six felt) were tested for pressure drop in a typical respirator cartridge to determine if they were acceptably breathable based on the National Institute for Occupational Safety and Health (NIOSH) requirement for respirator certification on breathing resistance (i.e., maximum initial inhalation resistance of 40 mm of water (mmH_2_O) for chemical-cartridge respirators for gases and/or vapors) [27]. Results showed the ACF felt was the more appropriate adsorbent material in ACF respirator cartridges with acceptable pressure resistance (<40 mmH_2_O) compared to ACF cloth, which exceeded two times the maximum limit. This may be attributable to the denser weaving of fibers. This indicates that using 100% ACF cloth in cartridges is not recommended and may result in respirators through which users may have difficulty in breathing [28]. Moreover, the ACF pressure drop study showed that combining two ACF types in a cartridge, including ACF cloth and felt combinations, resulted in acceptable pressure drop across these respirator components, demonstrating the possibility of incorporating dense ACF cloth with less dense ACF felt to achieve a compromise between adsorbent bulk density and permeability [28]. To further understand the potential application of ACFs in respiratory protection, it is important to investigate the adsorption characteristics of these ACF respirator cartridges, previously tested for acceptable breathing resistance, to determine their service lives for certain airborne pollutants.

Personal protective equipment (PPE), such as respirators, plays an important role in worker protection, particularly in situations wherein engineering and administrative control strategies are not feasible or sufficient. Thus, it is essential that respirators are continually improved in terms of design, comfort, and efficiency. With ACF showing promise as an adsorbent in respirators, more information on its suitability and limitations as an alternative adsorbent is needed. The purpose of this pilot study was to determine the adsorption characteristics, particularly breakthrough times and adsorption capacity, of commercially available ACFs of varying properties and compositions in respirator cartridges for toluene protection. It is hypothesized that the properties (e.g., density and surface area) and composition (e.g., single type and combination) of ACF in respirator cartridges will affect the adsorption characteristics of ACF for toluene. Findings of this study will be used to eventually achieve the ultimate goal of designing breathable and lighter ACF respirators that will provide sufficient worker protection against airborne contaminants.

## 2. Materials and Methods

### 2.1. Materials

Seven types of ACF materials (American Technical Trading, Inc., Pleasantville, NY, USA) with varying forms (1 cloth type (AC), 6 felt types (AF)), thickness, and density were tested. The ACFs were cut into 3 inch diameter discs and treated overnight prior to testing in a laboratory oven (Precision Compact Model 665, Thermo Scientific, Marietta, OH, USA) at 200 °C to desorb moisture and volatile impurities on the adsorbents. The ACF discs were placed in a desiccator to cool for a short period and then weighed (mg) using an analytical balance (Voyager Pro, Ohaus Corp., Parsippany, NJ, USA). The ACF discs were then placed and sealed in a typical respirator cartridge (7.62 cm (3 inch) internal diameter, 2.54 cm (1-inch) bed depth). Cartridges were filled with either 100% of each AF type or an AC/AF combination with pressure drop values of <40 mmH_2_O based on previous pilot study findings on pressure drop [28], resulting in 6 single AF cartridge (CS1-CS6) and 1 AC/AF combination cartridge (CC7) compositions (Table 1). The number of layers per ACF composition that were placed to fill the cartridge ranged from 5 to 11 depending on the ACF layer thickness (Table 1), and was determined in such a manner that the ACF materials were not excessively compressed in the cartridge, and thus maintained their natural density. The layer thickness for each ACF type was measured using a caliper. The volume of an ACF disc layer (cm^3^) was calculated by multiplying the disc area (45.6 cm^2^) by its layer thickness (cm). The density of each ACF type (g/cm^3^) was calculated by dividing the mass (g) of an ACF disc layer by its volume (cm^3^). The average ACF mass per cartridge type ranged from 3.96 ± 0.20 g (CS4 cartridge) to 8.01 ± 0.31 g (CS6 cartridge). There was a significant difference in average ACF mass (F(6,34) = 146.101, *p* < 0.01) between ACF cartridge types. Polypropylene sheets (Pall Life Sciences, Port Washington, NY, USA), cut into 7.62 cm (3 inch) diameter discs, were used to sandwich the bed of ACF discs in the cartridge, with the purpose of preventing the ACF material from potential shedding. Each ACF respirator cartridge was weighed again immediately before it was placed in the test chamber for chemical exposure to obtain the “pre-exposure” weight (mg). 

### 2.2. Characterization of ACF Structural Properties

The structural properties of the ACF materials, including surface area and porosity, were determined by nitrogen adsorption at 77 K using a Micromeritics ASAP 2020 automatic physisorption analyzer (Micromeritics Corp., Norcross, GA, USA), using a coverage factor, k = 2. The specific surface area was determined using the Brunauer–Emmett–Teller (BET) method based on the Rouquerol criterion, wherein a linear fit between 0.01–<0.2 relative pressure (P/P_0_) was obtained. Micropore volume was determined using the t-plot, and pore size distribution was obtained using the Barrett–Joyner–Halenda (BJH) method. Each ACF type was analyzed in triplicate (*n* = 3), resulting in a total of 21 physisorption analyses. The fiber organization and morphology of the ACF materials were also examined under scanning electron microscopy (SEM, FEI Quanta 200, Hillsboro, OR, USA). SEM images of each ACF type were obtained at 200× magnification to visualize the fiber organization in the ACF samples. 

### 2.3. Breakthrough Determination

ACF respirator cartridges were challenged with different concentrations of toluene in a customized cylindrical PTFE test chamber, with a 15.24 cm (6 inch) internal diameter. Laboratory-grade toluene (≥99.5%, Sigma-Aldrich, St. Louis, MO, USA) was used as the adsorbate without further purification. Toluene was used as the representative VOC in this study as similarly undertaken in several adsorption studies that tested ACF materials [15,18,19,20,22]. Toluene is among the widely used VOCs in occupational settings and a major organic vapor in indoor spaces [29]. Five toluene challenge concentrations (20, 100, 200, 300, and 500 ppm) were used, most of which represent various occupational exposure limits for toluene (from TLV-TWA (Threshold Limit Value − Time Weighted Average) = 20 ppm to IDLH (Immediately Dangerous to Life and Health) concentration = 500 ppm). 

Using a programmable syringe pump (Aladdin-1000, World Precision Instruments, Sarasota, FL, USA), liquid toluene was continuously injected at a specific rate into pre-conditioned air at constant temperature (23 °C), relative humidity (50%), and air flow (32 LPM) to create the desired toluene challenge concentration. The constant temperature and relative humidity used in this study were also used in previous ACF adsorption studies and in NIOSH service life testing of organic vapor respirators as standard experimental conditions [19,22,30]. This study tested a respirator cartridge type that is typically utilized in pairs in a dual-cartridge respirator, wherein each cartridge receives half of the total airflow. Given that one cartridge was tested at a time, the airflow used in the breakthrough tests was half (32 LPM) of the constant airflow used by NIOSH (64 LPM) in service life testing of respirator chemical cartridges (condition as received) for organic vapors for certification [30]. Dry, oil-free air was pre-conditioned and flow-controlled using a Miller-Nelson Model HCS-401 instrument (Assay Technology, Inc., Livermore, CA, USA). The air flow was measured downstream of the experimental setup before and after each day the breakthrough experiments were conducted using a Gilibrator 2 primary flow calibrator (Sensidyne, St. Petersburg, FL, USA). The residence time, RT = 217 msec, was calculated using RT = D*A/Q, wherein D (2.54 cm) is the cartridge bed depth, A (45.6 cm^2^) is the cartridge cross-section area, and Q (0.53 cm^3^/msec) is the flow rate [31]. Monitoring of the temperature and relative humidity in the test chamber was conducted using a temperature and relative humidity datalogger (HOBO Model U14-002, Onset Computer Corp., Pocasset, MA, USA). Breakthrough curves of toluene were obtained for each ACF cartridge configuration at each toluene concentration by continuous monitoring of the effluent (i.e., downstream of the ACF cartridge) using a photoionization detector (PID) (VOC-TRAQ II, MOCON Baseline Series, Lyons, CO, USA). The effluent was monitored continuously until 100% breakthrough occurred. The influent (i.e., upstream of the ACF cartridge) was also monitored continuously in the same manner using another VOC-TRAQ II PID to confirm the influent gas concentration. The exposure system was calibrated at the desired challenge concentration before and after every exposure run while using the PID units in the same manner as in the breakthrough experiments. The schematic diagram of the experimental setup for breakthrough determination is shown in Figure 1. 

The time in minutes when C_x_/C_0_ = 0.1 (i.e., 10% breakthrough time) and C_x_/C_0_ = 0.5 (i.e., 50% breakthrough time), wherein C_x_ = effluent or exit concentration and C_0_ = inlet or initial challenge concentration, were determined for each breakthrough curve, and compared for each ACF cartridge tested. The breakthrough times until C_x_ = 5 ppm (i.e., 5 ppm breakthrough time) were also determined and used in this study to estimate the service life of the ACF respirator cartridges, which is based on the NIOSH standard testing procedure for organic vapor cartridge service life [30]. Duplicates were run on 20% of the total breakthrough tests. Immediately after the toluene exposure, each ACF respirator cartridge was weighed using an analytical balance to obtain the “post-exposure” weight (mg).

The mass transfer zone (MTZ) length (L_T_) at five toluene concentrations for all ACF cartridge types was calculated to examine the diffusional constraints within each ACF cartridge using Equation (1):(1)$$L_{T}=\frac{t_{f}-t_{r}}{t_{f}}\times L$$
where t_f_ is the total equilibrium time (minutes) when C_x_/C_0_ = 1, t_r_ is the bed saturation time (minutes) when C_x_/C_0_ = 0.05, and L is the bed length (cm). 

### 2.4. Determination of Adsorption Capacity

Adsorption capacity is defined in this study as the amount of toluene adsorbed per mass of ACF material. The mass of toluene adsorbed on each ACF cartridge (mg) was obtained by subtracting the ACF pre-exposure weight (W_1_, mg) from the ACF post-exposure weight (W_2_, mg). The adsorption capacity (mg/g) of each ACF cartridge was calculated by determining the mass ratio of adsorbed chemical (mg) to the ACF materials (i.e., W_1_, g) in the cartridge as follows:(2)$$\mathrm{Adsorption}\;\mathrm{capacity}\;\left(\frac{\mathrm{mg}}{g}\right)=\frac{W_{2}\left(\mathrm{mg}\right)-{\;W}_{1}\left(\mathrm{mg}\right)}{W_{1}\left(g\right)}$$

### 2.5. Comparison of Adsorption Characteristics by ACF Type

The breakthrough times and adsorption capacities were compared by ACF cartridge types using the analysis of covariance (ANCOVA), which was conducted to determine significant differences in breakthrough times and adsorption capacities among the cartridge types when adjusted for toluene concentration. The breakthrough time (10%, 50%, or 5 ppm) for all toluene concentrations were pooled together for each cartridge type to calculate the adjusted mean breakthrough times, which were compared among the cartridge types. The same analysis was conducted to compare the adsorption capacity among the cartridge types. Data analysis was conducted using SPSS software version 25 (IBM, Armonk, NY, USA). *p* < 0.05 was considered as statistically significant.

## 3. Results and Discussion

### 3.1. ACF Structural Properties

The structural properties of the ACF materials are summarized in Table 2, including weave type, thickness, density, BET surface area, and microporosity. The non-woven AF types are 2- to 4.5-fold thicker than the woven AC type but are less dense (0.04–0.07 g/cm^3^) than the AC type (0.11 g/cm^3^), giving the AF types a spongy appearance. The BET surface area of the ACF materials ranged from 1204 to 2028 m^2^/g, and the micropore area ranged from 1122 to 1757 m^2^/g. The average % micropore by area ranged from 83.9 to 93.2% and the average pore size ranged from 1.58 to 1.69 nm (<2 nm), demonstrating that the ACF materials are mainly microporous. The BET surface area had a strong positive correlation with the micropore area (r = 0.997) but had a strong negative correlation with % micropore by area (r = −0.982). The organization of fibers in the ACF materials is shown in the SEM images. At 200× magnification, the AF types (AF1–AF6) show loose non-woven fibers that are randomly arranged, whereas the AC type (AC1) shows tightly woven fibers that give a denser appearance (Figure 2).

### 3.2. Breakthrough Characteristics by Cartridge Type

A total of 42 breakthrough curves were obtained in this study. Figure 3 shows the breakthrough curves for each cartridge type at five challenge concentrations of toluene, demonstrating varying breakthrough times. CS2 and CS6 cartridges that contain AF2 and AF6 (i.e., the two densest ACF felt types), respectively, have the longest breakthrough times (e.g., 10% breakthrough times of 809 and 873 min, respectively, at 20 ppm, and 44 and 43 min, respectively, at 500 ppm). In contrast, the CS1 cartridge that contains AF1, which is the least dense among the ACF types, has the shortest breakthrough time (e.g., 22 min for 10% breakthrough at 500 ppm). Moreover, CS2 and CS6 cartridges also have the highest average ACF mass (i.e., 7.76 and 8.01 g, respectively). The 10%, 50%, and 5 ppm breakthrough times have strong positive correlations with ACF density (r = 0.953, 0.948, and 0.960, respectively) and total ACF mass (r = 0.885, 0.911, and 0.903, respectively). The increase in toluene concentration for the CS6 cartridge occurs at a slower rate (e.g., 22.3 ppm/min at 500 ppm) from the breakpoint (i.e., point of the breakthrough curve wherein the effluent starts rising) compared to the other cartridge types (e.g., 40.8–53.7 ppm/min at 500 ppm), demonstrating a less steep slope of its breakthrough curve (Figure 3). Compared to other ACF properties, 10%, 50%, and 5 ppm breakthrough times have weak correlations with BET surface area (r = 0.286, 0.331, and 0.328, respectively), micropore area (r = 0.237, 0.283, and 0.280, respectively), % micropore by area (r = 0.461, 0.495, and 0.495, respectively), micropore volume (r = 0.273, 0.323, and 0.316, respectively) and pore size (r = 0.273, 0.161, and 0.266, respectively). 

The CC7 cartridge that contains the AF1/AC1 combination has the 2nd shortest breakthrough time, following CS1 (Figure 3). The purpose of incorporating a few layers of AC1 between the AF1 layers in the CC7 cartridge type is to increase the breakthrough time compared to using 100% AF1 in the CS1 cartridge type. Because CC7 has a longer breakthrough time (e.g., 6 min more in 10% breakthrough time at 500 ppm toluene) than CS1, combining the ACF cloth and felt may have improved toluene adsorption but not significantly enough to be comparable to cartridge types with dense ACF felt types (i.e., CS2 and CS6).

### 3.3. Effects of Toluene Concentration on Breakthrough Times

The effect of toluene concentration on the breakthrough curves was investigated in this study. Figure 4 demonstrates the breakthrough curves at five toluene concentrations as represented by the CS1 and CS6 cartridges. As the toluene concentration decreases, the breakthrough time of toluene through the ACF cartridge increases, regardless of the differences in ACF density in the cartridges (Figure 4). Such a concentration effect is expected because lower toluene concentration means fewer toluene molecules adsorb onto the ACF and move through the ACF bed depth, thus taking longer for breakthrough to occur.

Such an increase in breakthrough time, however, is not linear with the toluene concentration. The breakthrough time increase at lower toluene concentrations was observed to be greater than that at higher concentrations, even at the same proportional change in toluene concentration. For example, the breakthrough time increase from 100 to 300 ppm toluene is larger compared to the time increase from 300 to 500 ppm toluene for CS1 and CS6 cartridge types, despite the same concentration increment of 200 ppm (Figure 4). Such a non-linear relationship between breakthrough times (10%, 50%, and 5 ppm) and toluene concentration was found for all cartridge types, as shown in Figure 5. Similar results were found in a previous ACF study that involved the testing of ACF samples against toluene for respirator application [26]. This finding indicates that the ACF cartridges may be more efficient in providing a longer service life at lower toluene concentrations, and may be attributed to the microporosity of the ACF adsorbents, because micropores in activated carbons are mainly responsible for the enhanced adsorption of vapors at low ambient concentrations and are favorably filled at low concentrations due to the overlapping of attractive forces associated with the close proximity of the opposite pore walls [32,33,34]. A study by Fuertes et al. [35] found that the adsorbate amount at low concentrations depends on the pore size distribution of the adsorbent, specifically, its microporosity, which allows higher adsorption capacities at low concentrations. Evidence has shown the importance of narrow microporosity in ACF materials in optimizing VOC adsorption at low concentrations [35].

### 3.4. Breakthrough Times by Cartridge Type

The 10%, 50%, and 5 ppm breakthrough times were determined and compared among cartridge types, with the aim of determining the potential respiratory protection each cartridge type may provide at different concentrations of toluene. Figure 6 shows the average breakthrough times, which were obtained by pooling the breakthrough times for all toluene concentrations, for all cartridge types. The CS6 cartridge has the highest 10%, 50%, and 5 ppm breakthrough times (267, 321, and 272 min, respectively), followed by CS2 (244, 271, and 251 min, respectively). The ACF types in the CS6 and CS2 cartridges (AF6 and AF2, respectively) are the densest among the felt types, which may have contributed to the high breakthrough times. However, there was no significant difference in 10% (F(6,27) = 0.308, *p* = 0.927), 50% (F(6,27) = 0.357, *p* = 0.899), and 5 ppm (F(6,27) = 0.284, *p* = 0.939) breakthrough times between ACF cartridge types, while adjusting for toluene concentration. In practical applications, however, a 100 min difference in the duration of worker protection time still matters. For example, the use of the CS6 cartridge would still be recommended over the use of the CS1 cartridge due to the approximately 130 min difference in the average 5 ppm breakthrough times. 

### 3.5. Estimation of ACF Cartridge Service Lives

The breakthrough time to reach 5 ppm toluene (i.e., 5 ppm breakthrough time) was used in this study to estimate the service lives of the ACF cartridges. This is based on the NIOSH standard testing procedure for organic vapor chemical cartridges challenged with a 1000 ppm carbon tetrachloride, wherein the maximum allowable breakthrough is set at 5 ppm with a minimum service life requirement of 50 min [30]. Figure 7 shows the 5 ppm breakthrough times for all toluene challenge concentrations by cartridge type. For example, the ACF cartridges will take 21–44 min to reach 5 ppm breakthrough at 500 ppm toluene exposure (i.e., IDLH value for toluene) and will take 51 min–1.5 h at 200 ppm toluene exposure (i.e., OSHA PEL-TWA for toluene) (Figure 7). At the lowest toluene concentration of 20 ppm, 5 ppm breakthrough time for the ACF cartridges ranges from 8.5 to 16 h (Figure 7). It is important to note that these breakthrough times are obtained by challenging one cartridge only. Thus, the use of a typical dual-cartridge respirator may provide twice the protection time observed in this study. 

The service life of an organic vapor cartridge at other conditions (e.g., different challenge concentrations) may be estimated by various methods, such as breakthrough experimental tests using full-size cartridges or modified containers [36,37,38] and predictive modeling [39,40]. However, one rule of thumb for estimating such service life states that reducing the challenge concentration by a factor of 10 will increase the service life of a cartridge by a factor of five [41]. Using this rule of thumb, three concentration values (1000 ppm as the reference challenge concentration, 100 ppm, and 10 ppm) and their corresponding minimum service life requirements (50, 250, and 1250 min, respectively) were used to determine the best-fitting equation, y = 6250x^−0.699^, to calculate the minimum service life requirements (y) for the five toluene concentrations (x) used in this study. The calculated minimum service life requirements were then compared to the range of experimental 5 ppm breakthrough times obtained for all ACF cartridges (Table 3). Among the toluene challenge concentrations, only 20 ppm had values within its range of 5 ppm breakthrough times (i.e., 770–960 min) that met the minimum service life requirement (770 min). Specifically, the CS6 and CS2 cartridges met the minimum service life requirement, with breakthrough times at 960 and 857 min, respectively. The remainder of the concentrations had maximum 5 ppm breakthrough times that were below the minimum service life requirements. This finding again supports the earlier discussion that indicates the better performance and suitability of the ACF cartridges in providing longer respiratory protection at lower toluene concentrations [26].

### 3.6. Mass Transfer Zone (MTZ) Length by Cartridge Type

Table 4 shows the 5% breakthrough time (t_r_), the total equilibrium time (t_f_), the mass transfer zone (MTZ) length (L_T_), and L_T_/L. The CS3 cartridge showed the lowest to the second lowest L_T_ (0.19–0.25) or L_T_/L (0.07–0.10) throughout the five concentration conditions, indicating a more effective cartridge among the tested cartridges [42,43]. On the other hand, some of the highest L_T_ and L_T_/L values were shown in the CS5 (0.81–1.11 and 0.32–0.44 cm, respectively) and CS6 (0.45–0.89 and 0.18–0.35 cm, respectively) cartridges throughout most of the five toluene concentrations. The higher LT means that the diffusion of toluene molecules will be more restricted within the adsorbent, which is more likely to result in a decrease in the breakthrough adsorption capacity [42,43]. However, our findings showed a weak correlation between the LT and adsorption capacity (r = 0.010 to 0.474), regardless of the toluene concentration.

### 3.7. Adsorption Capacity by Cartridge Type

Figure 8 shows the average adsorption capacity values, which were obtained by pooling the adsorption capacities for all toluene concentrations, for all cartridge types. The CS4 cartridge has the highest adsorption capacity (467 mg/g), followed by the CS5 cartridge (450 mg/g). The ACF types in the CS4 and CS5 cartridges (AF4 and AF5, respectively) have the highest measured BET surface areas among all ACF types, which may be attributed to the high adsorption capacities. The CS1 cartridge has the lowest adsorption capacity, which may be due to its ACF type, AF1, which has the lowest surface area. Adsorption capacity has a strong positive correlation with BET surface area (r = 0.980), micropore area (r = 0.990), and micropore volume (r = 0.986). Increasing surface area of either felt or cloth forms of ACF has been shown to increase adsorption capacity for toluene, which is attributed to the increasing amount of available adsorption sites for toluene molecules at a given ACF mass [2]. Previous studies on ACF and other carbon adsorbents challenged with other organic compounds have shown a positive linear relationship between the adsorption capacity and the surface area, independent of the adsorbent type. For example, the adsorption capacity of ACF increased for nonafluorobutyl methyl ether (NFE) as the ACF specific surface area increased from 918 to 1754 m^2^/g [44]. Similarly, on a mass basis, the adsorption capacities of ACF and GAC (with BET surface area of 949 and 706 m^2^/g, respectively) for aromatic organic compounds were found to be higher than those of carbon nanotubes (with BET surface area from 164 to 537 m^2^/g) [45]. Moreover, there was a significant difference in adsorption capacity (F(6,27) = 13.736, *p* < 0.01) between ACF cartridge types, when adjusted for toluene concentration. Compared to other ACF properties, adsorption capacity has a strong negative correlation with % micropore by area (r = −0.905) but has weak correlations with density (r = 0.369) and pore size (r = 0.045).

Pairwise comparison of the average adsorption capacity between all 21 possible pairs of cartridge types was conducted. Ten of the 21 pairwise comparisons (47.6%) showed significance differences, with *p*-values ranging from <0.001 to 0.047. Figure 8 shows whether or not there is significant difference in adsorption capacity between each cartridge pair.

Figure 9 shows the adsorption capacities of the ACF cartridge types by toluene concentration, ranging from 252 to 532 mg/g. The adsorption capacity shows an increasing trend as the challenge toluene concentration increases for all cartridge types (Figure 9). However, the rate of increase varies by cartridge type. The cartridges filled with ACF types with the highest surface areas (i.e., CS4 and CS5 cartridges) have the highest increase rate as demonstrated by their steepest slopes, whereas the cartridge filled with ACF with the lowest surface area (i.e., CS1 cartridge) has the lowest increase rate. These results are similar to those in our previous ACF study, which involved testing of ACF samples that are not in cartridges [2]. 

The adsorption isotherms of toluene by ACF cartridge type were derived using the calculated adsorption capacities and the five toluene concentrations used in this study. Each toluene concentration was converted into relative pressure (P/P_0_), where saturation pressure, P_0_ = 3.5 kPa at 23 °C. Figure 10 shows the adsorption isotherms of toluene at 23 °C for each ACF cartridge type over the range of relative pressure from 0.000 to 0.015. At this narrow relative pressure range, the shape of the adsorption isotherms for all ACF cartridge types demonstrates Type I or the Langmuir type of isotherms according to the IUPAC classification, indicating that the ACF materials are essentially microporous [32]. This is supported by the study findings that 83.90–93.23% of the ACF surface area are comprised of micropores (Table 2) and previous ACF adsorption studies [7,8,9,10].

## 4. Conclusions

Activated carbon fiber is considered to be an alternative adsorbent that may be used to develop thinner, lightweight, and efficient respirators, but its adsorption characteristics need to be investigated further. The findings of this study show the extent of respiratory protection that ACF respirator cartridges may provide against airborne toluene and, potentially, other VOCs. Cartridges containing the densest ACF felt types (i.e., AF2 and AF6) have the longest breakthrough times (e.g., ~250–270 min to 5 ppm breakthrough time), and thus provide the longest protection time against toluene. Cartridges containing ACF felt types with the highest specific surface areas have the highest adsorption capacity (~450–470 mg/g). Although the ACF cloth type was previously shown to have promising adsorption characteristics (i.e., high adsorption capacity and long breakthrough times), incorporating ACF cloth with ACF felt did not significantly increase the breakthrough time. Thus, there is no need to use ACF cloth in ACF cartridges to improve adsorption, given one of its disadvantages is high pressure drop. Using the appropriate ACF felt type in terms of density (~0.07 g/cm^3^) and specific surface area (~2000 m^2^/g) will be sufficient in ACF respirator cartridges.

The breakthrough times of toluene across the ACF cartridges decreased with increasing toluene challenge concentration, but this relationship was shown to be non-linear. The ACF cartridge may provide respiratory protection for <1 h for 500 pm toluene, and for 8–16 h for 20 ppm toluene. At its current configuration, the ACF cartridges may be more reliable for use at low ambient toluene concentrations, which may be attributable to the ACF’s microporosity. Nonetheless, findings indicate that ACF felt types still have potential to be used as short-term protection at high toluene concentration, such as during emergency escape or evacuation (e.g., high-volume spills resulting in high ambient concentration), but the design of the ACF respirator or respirator component needs to be modified. Given its non-granular and light fabric form, ACF may be utilized in respirators without the need for hard containment (e.g., cartridge) and be developed into a light N95-type filtering facepiece respirator (FFR) that may be potentially reusable. If used as such, the ACF felt may be utilized with a contact surface area that is larger than that of the ACF cartridge tested in this study (~46 cm^2^), which is possible given a typical N95 FFR surface area of approximately 165 cm^2^. However, increasing the current ACF bed depth (1 in) to increase protection time is not recommended because this may result in unacceptable breathing resistance. The ACF cartridges in this study have been previously tested and shown to have acceptable pressure drop based on NIOSH criteria for respirator use approval, but increasing the bed depth may compromise the breathability of the respirator. Further studies on the adsorption characteristics of improved ACF respirator and/or respirator components for toluene and other VOCs at various environmental (e.g., temperature and relative humidity) and worker (e.g., breathing pattern and rate) conditions, in addition to comparison studies with currently available respirator cartridges (i.e., using GAC as adsorbent), are warranted to better understand the strengths and limitations of ACF as an adsorbent for respiratory protection.

## Figures and Tables

**Figure 1 ijerph-18-08505-f001:**
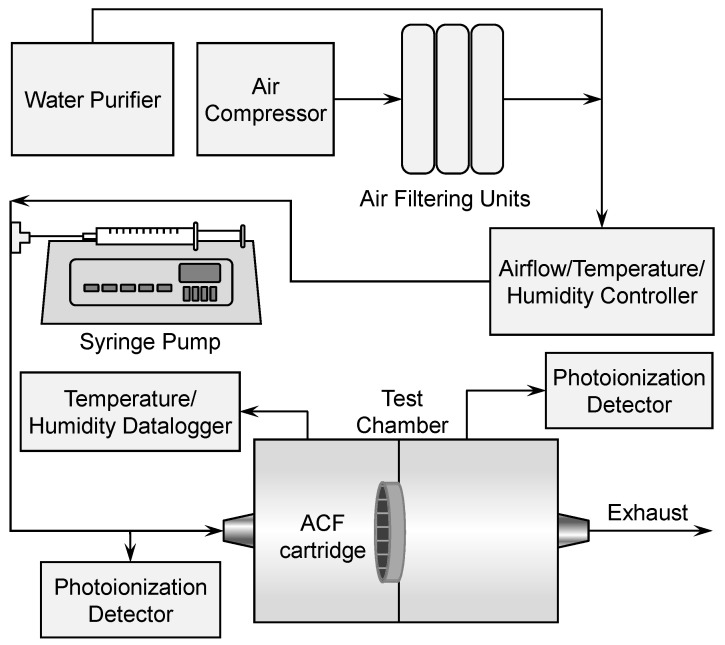
Experimental setup for breakthrough determination for toluene across an ACF respirator cartridge.

**Figure 2 ijerph-18-08505-f002:**
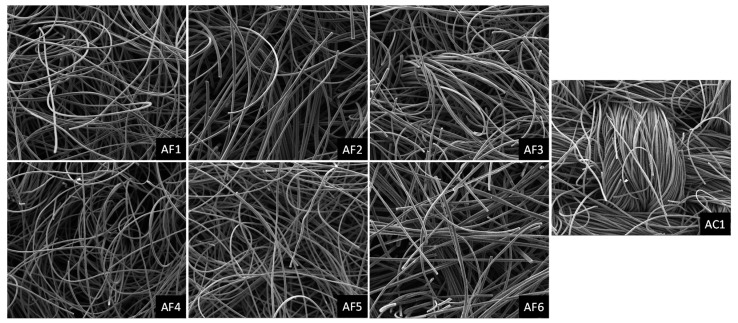
SEM images of activated carbon fibers (ACF) at 200× magnification. ACF felt types (AF1 to AF6); ACF cloth type (AC1).

**Figure 3 ijerph-18-08505-f003:**
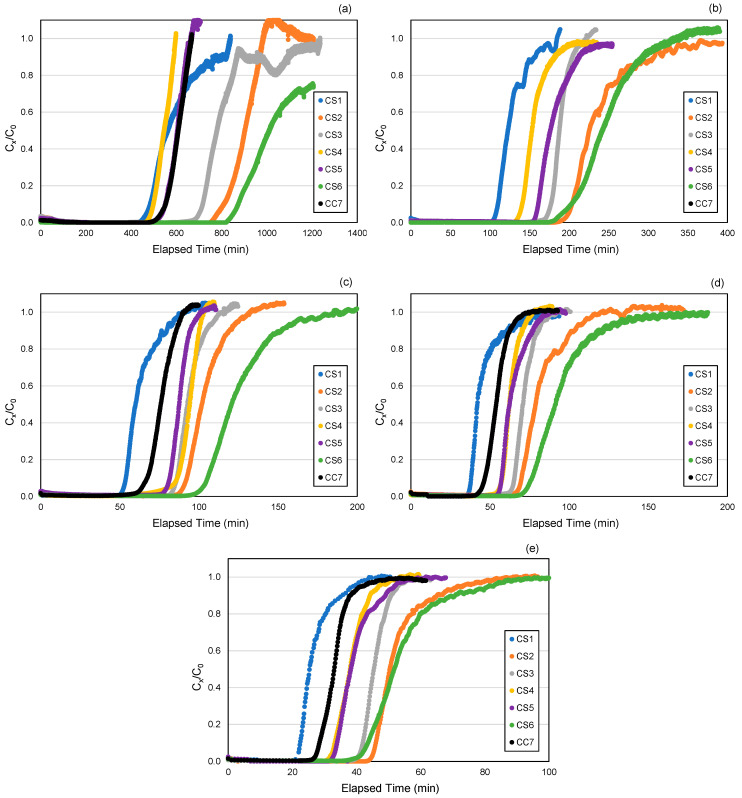
Breakthrough curves by cartridge type at 5 toluene concentrations: (**a**) 20 ppm, (**b**) 100 ppm, (**c**) 200 ppm, (**d**) 300 ppm, (**e**) 500 ppm. Cartridge type (ACF composition): CS1 (100% AF1); CS2 (100% AF2); CS3 (100% AF3); CS4 (100% AF4); CS5 (100% AF5); CS6 (100% AF6); CC7 (71% AF1/29% AC1).

**Figure 4 ijerph-18-08505-f004:**
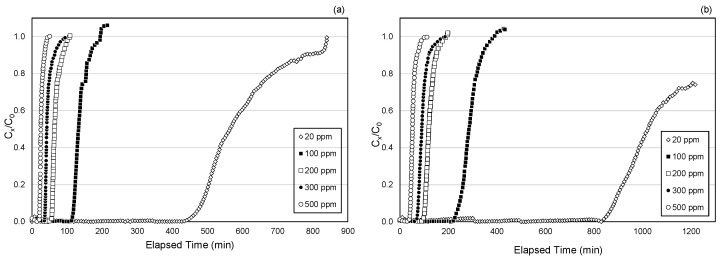
Breakthrough curves for (**a**) CS1 (100% AF1) and (**b**) CS6 (100% AF6) cartridge types by toluene concentration.

**Figure 5 ijerph-18-08505-f005:**
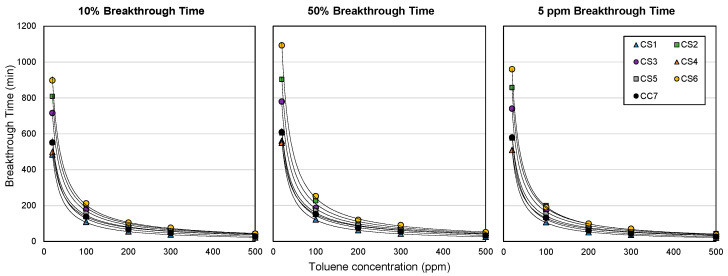
Breakthrough times as a function of toluene concentration by cartridge type. Cartridge type (ACF composition): CS1 (100% AF1); CS2 (100% AF2); CS3 (100% AF3); CS4 (100% AF4); CS5 (100% AF5); CS6 (100% AF6); CC7 (71% AF1/29% AC1).

**Figure 6 ijerph-18-08505-f006:**
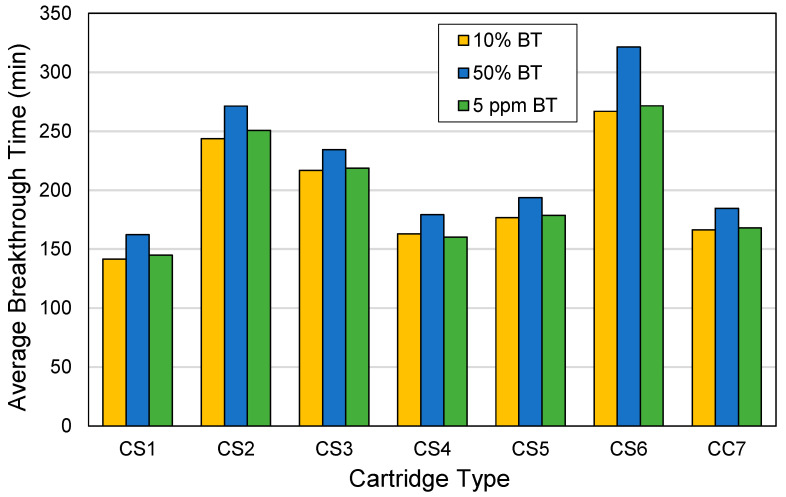
Average breakthrough times by cartridge type. Cartridge type (ACF composition): CS1 (100% AF1); CS2 (100% AF2); CS3 (100% AF3); CS4 (100% AF4); CS5 (100% AF5); CS6 (100% AF6); CC7 (71% AF1/29% AC1).

**Figure 7 ijerph-18-08505-f007:**
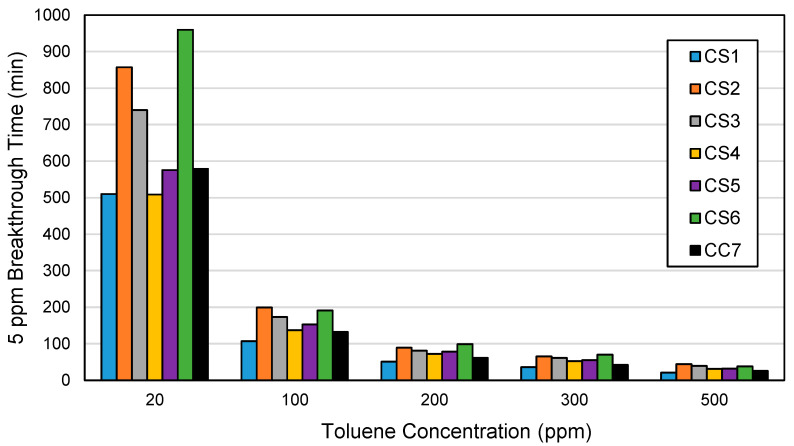
Toluene 5 ppm breakthrough time by cartridge type and toluene concentration. Cartridge type (ACF composition): CS1 (100% AF1); CS2 (100% AF2); CS3 (100% AF3); CS4 (100% AF4); CS5 (100% AF5); CS6 (100% AF6); CC7 (71% AF1/29% AC1).

**Figure 8 ijerph-18-08505-f008:**
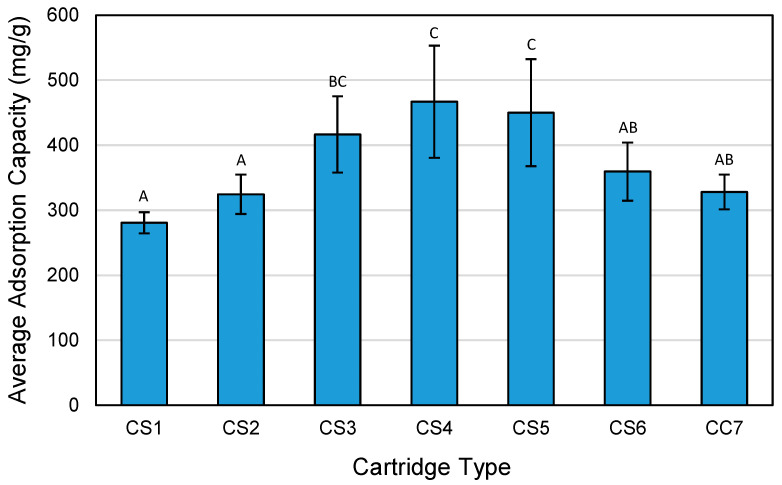
Average adsorption capacity by cartridge type. Shown with error bars. The letters above each bar represent differences or similarities between cartridge types. Cartridge types with the same letter are not significantly different by average comparison. Cartridge types with a different letter are significantly different by average comparison. Cartridge type (ACF composition): CS1 (100% AF1); CS2 (100% AF2); CS3 (100% AF3); CS4 (100% AF4); CS5 (100% AF5); CS6 (100% AF6); CC7 (71% AF1/29% AC1).

**Figure 9 ijerph-18-08505-f009:**
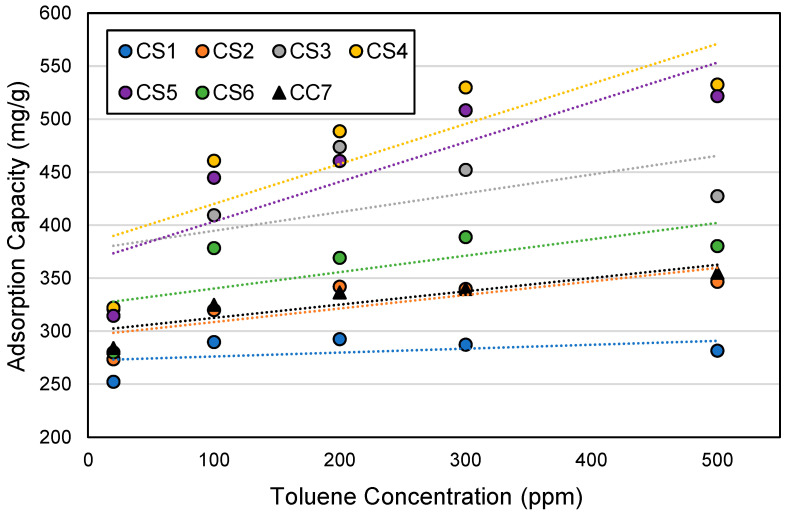
Adsorption capacity by toluene concentration and cartridge type. Cartridge type (ACF composition): CS1 (100% AF1); CS2 (100% AF2); CS3 (100% AF3); CS4 (100% AF4); CS5 (100% AF5); CS6 (100% AF6); CC7 (71% AF1/29% AC1).

**Figure 10 ijerph-18-08505-f010:**
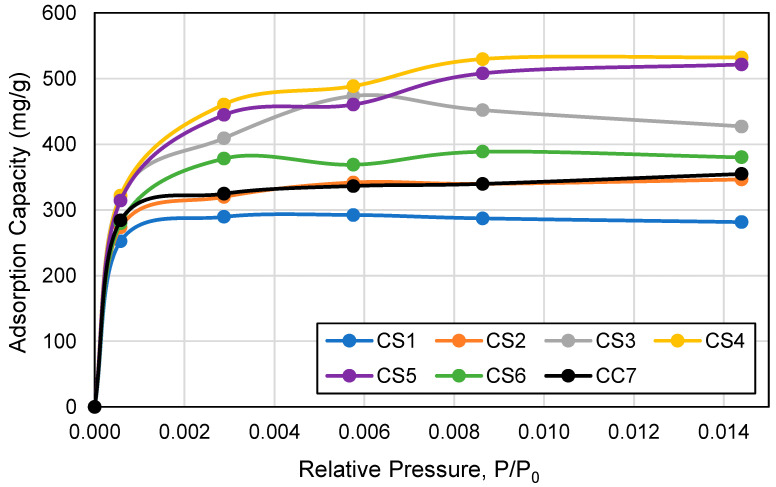
Toluene adsorption isotherms at 23 °C by cartridge type over the relative pressure (P/P0) range 0.000–0.015. Cartridge type (ACF composition): CS1 (100% AF1); CS2 (100% AF2); CS3 (100% AF3); CS4 (100% AF4); CS5 (100% AF5); CS6 (100% AF6); CC7 (71% AF1/29% AC1).

**Table 1 ijerph-18-08505-t001:** Activated carbon fiber (ACF) composition, number, and mass of layers by cartridge type.

Cartridge Type	ACF Composition ^a^ by % Weight	Number of ACF Layers	Average ACF Mass (g) ^b^
CS1	100% AF1	11	4.97 ± 0.06
CS2	100% AF2	9	7.76 ± 0.64
CS3	100% AF3	8	5.31 ± 0.46
CS4	100% AF4	10	3.96 ± 0.20
CS5	100% AF5	8	4.43 ± 0.16
CS6	100% AF6	5	8.01 ± 0.31
CC7	71% AF1/29% AC1	8 (AF1)/3 (AC1)	5.04 ± 0.06

^a^ AF1–AF6 (ACF felt types); AC1 (ACF cloth type). ^b^ Pre-exposure weight.

**Table 2 ijerph-18-08505-t002:** Activated carbon fiber (ACF) structural properties by ACF type.

ACF Type	Weave Type	Thickness (cm) ^a^	Density (g/cm^3^) ^a^	BET Surface Area (m^2^/g) ^b^	Micropore Area (m^2^/g) ^b^	% Micropore by Area ^b^	Micropore Volume (cm^3^/g) ^b^	Pore Size (nm) ^b^
AF1	Non-woven	0.23	0.043	1203.63 ± 2.61	1122.14 ± 1.66	93.23 ± 0.10	0.43 ± 0.00	1.66 ± 0.00
AF2	Non-woven	0.29	0.069	1265.48 ± 34.76	1172.93 ± 35.85	92.68 ± 0.29	0.45 ± 0.01	1.58 ± 0.00
AF3	Non-woven	0.29	0.057	1707.28 ± 19.45	1529.05 ± 14.38	89.56 ± 0.18	0.59 ± 0.01	1.59 ± 0.04
AF4	Non-woven	0.24	0.043	2091.13 ± 9.28	1757.11 ± 5.11	84.03 ± 0.19	0.69 ± 0.00	1.62 ± 0.00
AF5	Non-woven	0.30	0.041	2028.08 ± 12.65	1701.54 ± 9.49	83.90 ± 0.06	0.67 ± 0.00	1.69 ± 0.00
AF6	Non-woven	0.50	0.070	1474.60 ± 26.56	1350.12 ± 33.55	91.55 ± 0.62	0.51 ± 0.01	1.67 ± 0.00
AC1	Woven	0.11	0.101	1903.45 ± 0.56	1637.87 ± 0.15	86.05 ± 0.03	0.64 ± 0.00	1.59 ± 0.00

^a^ average of 2 samples (*n* = 2) [28]. ^b^ average of 3 samples (*n* = 3).

**Table 3 ijerph-18-08505-t003:** Minimum service life requirement and experimental 5 ppm breakthrough times per organic vapor cartridge by toluene concentration.

Toluene Concentration (ppm)	Minimum Service Life Requirement (Minutes) ^a^	5-ppm Breakthrough Times (Minutes) ^b^
500	81	21–38
300	116	36–70
200	154	51–99
100	250	107–191
20	770	510–960

^a^ Calculated based on the rule of thumb stating that reduction in concentration by a factor of 10 increases service life by a factor of 5 [41]. ^b^ Range obtained for all ACF cartridges tested.

**Table 4 ijerph-18-08505-t004:** Mass transfer zone (MTZ) properties from breakthrough curves by cartridge type and toluene concentration (bed length (L) = 2.54 cm, flow rate = 32 LPM).

MTZ Properties	Cartridge Type	C_0_ = 20 ppm	C_0_ = 100 ppm	C_0_ = 200 ppm	C_0_ = 200 ppm	C_0_ = 500 ppm
Length of mass transfer zone, L_T_ (cm)	CS1	0.42	0.31	1.15	0.30	0.30
	CS2	0.34	0.29	0.30	0.33	0.21
	CS3	0.25	0.19	0.20	0.22	0.23
	CS4	0.63	0.27	0.35	0.77	0.40
	CS5	0.28	0.81	0.85	0.25	1.11
	CS6	0.89	0.61	0.38	0.50	0.45
	CC7	0.32	0.33	0.37	0.42	0.46
L_T_/L	CS1	0.17	0.12	0.45	0.12	0.12
	CS2	0.14	0.12	0.12	0.13	0.08
	CS3	0.10	0.07	0.08	0.09	0.09
	CS4	0.25	0.10	0.14	0.30	0.16
	CS5	0.11	0.32	0.34	0.10	0.44
	CS6	0.35	0.24	0.15	0.20	0.18
	CC7	0.13	0.13	0.15	0.17	0.18
Total equilibrium time, t_f_ (minute)	CS1	561	122	95	42	25
	CS2	903	225	102	78	49
	CS3	779	187	91	70	45
	CS4	648	153	94	80.5	38
	CS5	604	225	119	62	58.5
	CS6	1315	252	120	91	51
	CC7	609	152	75	54	33
5% breakthrough time, t_r_ (minutes)	CS1	468	107	52	37	22
	CS2	781	199	90	68	45
	CS3	701	173	84	64	41
	CS4	488	137	81	56	32
	CS5	537	153	79	56	33
	CS6	852	191	102	73	42
	CC7	532	132	64	45	27

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
