# Peer review of "Adsorption Characteristics of Activated Carbon Fibers in Respirator Cartridges for Toluene"

_ijerph, 2021, doi:10.3390/ijerph18168505_

Round 1
Reviewer 1 Report
the proposed work is concise, well presented and resulted form numerous experiments and data analysis. However, major revisions should be taken into account. This required modifications are resumed in the attached document.

Reviewer 2 Report
Manuscript Number: ijerph-1302575
|
|
|
Minor drawbacks and recommended improvements |
|
1 |
Line 28, page 1. |
The Keywords activated carbon fibers; adsorption; respirator and cartridges are in the title. |
|
2 |
Line 219, page 6. |
The authors should show these results of 42 breakthrough curves in the figure. |
|
3 |
Line 228, page 7. |
The authors should indicate reference the other studies with other cartridge types. |
|
4 |
Figure 3, 4, 5, 6, 7, 8 and 9. |
The description for each cartridge type should be in the title of these figures. |
|
5 |
Line 307, page 14. |
The authors should indicate which methods and the references. |
|
6 |
Line 334, page 11. |
The authors should indicate the type of organic compounds (adsorbate) and discuss these results (positive linear relationship between the adsorption capacity and the surface area, independent of the adsorbent type) in relation to the properties of the adsorbate. |
|
7 |
Line 189, page 5. |
The authors should indicate the mathematical equation to calculate the adsorption capacity |
|
8 |
Line 389-391, page 13. |
In the conclusion section, there is no reference, this was previously indicated in section 3. |
Reviewer 3 Report
Review:
Adsorption Characteristics of Activated Carbon Fibers in Respirator Cartridges for Toluene
This paper is on adsorption studies of ACF to see its suitability as respirator cartridge for Toluene. This paper cannot be published in the current form and needs extensive review before considering publication in IJERPH.
My specific concerns are
- The abstract need to rewrite to include the need of the work with the current context of the issue. There are some mistakes in the abstract. They should have one-way to describe breakthrough times not two like in the abstract ( Breakthrough times (10%, 50% and 5-ppm) and breakthrough times (e.g., ~250-270 min to 5-ppm BT).
- Why Toluene is chosen as a system, for adsorption need explanation.
- NIOSH requirement has to be referenced and cited well to get the understanding.
- We do see a variation in the goal of the paper, what is the goal of the paper? And this should be same all through.
- Why not a 100% cloth type ACF is not compared? In Table 1 and this is one of the comparison to be done.
- Is there any correlation between thickness and BET surface area, I do not see one in Table 2, and the question is why?
- There should be more analysis with Breakthrough time with each of the parameters of Table 2.
- There is a huge temperature dependency in this process and this is not included in this study, which is to be included.
- Same reasoning applies for Humidity.
- Conclusion does not justify the abstract and introduction and it is not supported with enough results.
Round 2
Reviewer 3 Report
The authors have responded to my queries, and I am satisfied with the answers.